# Moiré Graph Transformer: Eliminating Graph Positional Encoding with Focused Attention

## Abstract

Graph neural networks (GNNs) have increasingly adopted transformer architectures to capture long-range dependencies. However, integrating structural information into graph transformers remains challenging, often necessitating complex positional encodings or masking strategies. In this paper, we propose the Moiré Graph Transformer (MoiréGT), which introduces a novel focused attention mechanism that eliminates the need for explicit graph positional encodings. Our model effectively captures structural context without additional encodings or masks by adjusting attention scores based on a learnable focus function of node distances. We theoretically demonstrate that multiple attention heads with different focus parameters can implicitly encode positional information akin to moiré patterns. Experiments on 3D molecular graphs show that MoiréGT achieves significant performance gains over state-of-the-art models on the QM9 and PCQM4Mv2 datasets. Additionally, our model achieves competitive results on 2D graph tasks, highlighting its versatility and effectiveness.

## 1 Introduction

Graph neural networks (GNNs) have incorporated the transformer architecture, leveraging its success in other domains (Ying et al., 2021). Graph transformers (GTs) demonstrate strong performance on tasks involving node relationships (Dwivedi & Bresson, 2020), primarily due to their global attention mechanism. In graph contexts, the permutation invariance of attention is advantageous, as graphs are concerned with topological relationships and are invariant under the $SE(3)$ group (Bronstein et al., 2021).

However, GTs that rely solely on global attention without incorporating structural adjustments often perform poorly (Kreuzer et al., 2021). This is similar to how fully connected layers can fail when lacking inductive biases. Incorporating graph structural information is crucial for effective learning. While positional encodings have been employed to inject structural information (Dwivedi et al., 2021), designing encodings that uniquely and meaningfully represent topological positions while preserving invariance properties is challenging.

Masking techniques can hide irrelevant nodes and implicitly convey structural information, but it's limiting the global receptive field of attention mechanisms. This leads us to ask: *Can we develop a method that leverages attention while preserving structural context, effectively combining the benefits of positional encoding and masking?*

We introduce the **Moiré Graph Transformer (MoiréGT)**, a graph transformer with a **focused attention mechanism**. This allows the model to focus on specific ranges of nodes relative to nodes of interest by adjusting attention scores within the self-attention layer. We use a non-linear focus function with two learnable parameters—shift and width—to emphasize nodes within the focus area while filtering out others. Our theoretical analysis shows that moiré patterns with phase distortions can effectively encode positional information.

Experimentally, we demonstrate **significant performance gains on 3D graphs—QM9 and PCQM4Mv2**—and we also tested our model on 2D graphs to show its versatility.

**Our key contributions are as follows:**

- We introduce the focus mechanism, an explicit filtering mechanism based on spatial relationships, providing a strong inductive bias for graph learning tasks.
- Our model eliminates the need for a virtual node, reducing computational overhead.
- Extensive experiments demonstrate the efficiency and effectiveness of our model, achieving state-of-the-art results on challenging benchmarks with fewer parameters than competing models.

## 2 RELATED WORKS

**Graphormer** (Ying et al., 2021) adds centrality encoding into node features and employs learnable spatial and edge encodings to adjust attention values. Unlike Graphormer, we filter and emphasize node features solely based on topological distance, providing a simpler and more distinguishable method for each node.

**GraphGPS** and **TokenGT** (Rampášek et al., 2022; Kim et al., 2022) utilize orthonormal graph Laplacian eigenvectors to represent nodes' topological locations, which is not suitable for transductive graph learning tasks due to the need for eigen decomposition of the adjacency matrix and large input feature dimensions. Our method does not require eigendecomposition and can use any node features.

**GRPE** (Park et al., 2022) employs learnable positional encodings derived from the shortest path distance to adjust attention scores, a subset of the approach used in Transformer-M.

**Transformer-M** (Luo et al., 2022) incorporates degree encodings and sums of 3D distance encodings added to the input node features. It also uses shortest path distance, edge, and 3D distance encodings within attention. Despite their complex structure, they did not use this information for filtering as we do, leading to larger parameter numbers without clear performance gains from each component.

**EGT** (Hussain et al., 2022) adjusts global attention using edge embeddings, effectively inferring edge data. However, without filtering, their global attention lacked sufficient inductive bias, resulting in models 5 to 10 times larger than others and requiring clipping to stabilize training.

Moreover, these models often require a virtual node to make predictions, adding computational overhead due to its global connectivity. In contrast, our model does not require a virtual node.

## 3 METHOD

### 3.1 FOCUSED ATTENTION MECHANISM

The standard attention mechanism with global attention is formulated as follows:

$$\text{Attention}(Q, K, V) = \text{softmax}\left(\frac{QK^T}{\sqrt{d_k}}\right) V, \tag{1}$$

where $Q$, $K$, $V$ are the query, key, and value matrices derived from the node features, and $d_k$ is the dimension of the key vectors. While this global attention allows the model to capture relationships between any pair of nodes, it lacks inherent structural information about the graph topology, as it treats all nodes equally without considering their structural proximity.

To incorporate structural information, we propose a novel focused attention mechanism (equation 2) that adjusts the attention scores based on the distances between nodes.

$$\text{Attention}(Q, K, V) = \text{softmax}\left(\frac{QK^T}{\sqrt{d_k}}\right) f(D, \mu, \sigma) V \tag{2}$$

Where $f(D, \mu, \sigma)$ is a learnable focus function that emphasizes nodes within a certain distance range relative to each node of interest. The focus function depends on the distance matrix $D$ and has two learnable parameters—shift ($\mu$) and width ($\sigma$)—which allow the model to dynamically adjust its focus on the graph structure.

Our focus mechanism serves as a learnable filter on the attention scores, enabling the model to selectively emphasize or mask certain nodes based on their structural distance. This adjustment effectively incorporates structural information without the need for explicit positional encoding or an additional learnable projection matrix.

We compute and use a distance matrix $D$ when the dataset allows it. Each element $d_{ij} = \|c_i - c_j\|_2$, where $c_i$ and $c_j$ are the coordinates of nodes $i$ and $j$, respectively.

Each focus function has two learnable parameters: width ($\sigma$) and shift ($\mu$). The width parameter controls the spread or range of the focus function, adjusting how narrowly or broadly the model attends to nodes around $\mu$. The shift parameter adjusts the center of the focus function, effectively shifting the function along the distance axis.

To explore the effectiveness of our focus mechanism, we experimented with five different focus functions: Gaussian, Laplacian, Cauchy, Triangle, and MirroredSigmoid. Each function offers unique characteristics in how it adjusts the attention scores based on the distances between nodes. Figure 1 shows each focus function varies in form with different widths.

**Gaussian**: Defined as $f(d) = \exp\left(-\frac{(d-\mu)^2}{\sigma}\right)$, it provides a smooth, bell-shaped curve centered around $\mu$, emphasizing nodes at distances close to $\mu$.

**Laplacian**: Defined as $f(d) = \exp\left(-\frac{|d-\mu|}{\sigma}\right)$, it emphasizes nodes near $\mu$ with a sharper peak and heavier tails compared to the Gaussian.

**Cauchy**: Defined as $f(d) = \frac{1}{1+\left(\frac{d-\mu}{\sigma}\right)^2}$, it peaks at $d = \mu$ and decreases inversely with the square of the distance from $\mu$, maintaining attention to nodes even at larger distances.

**Triangle**: Defined as $f(d) = \max\left(1 - \frac{|d-\mu|}{\sigma}, 0\right)$, it creates a linear decrease in attention from $\mu$, reaching zero at $d = \mu \pm \sigma$.

**MirroredSigmoid**: Defined as $f(d) = \frac{1}{1+\exp\left(\frac{(d-\mu)}{\sigma}\right)}$, it produces an S-shaped curve, effectively modeling scenarios where attention is needed for nodes closer than $\mu$.

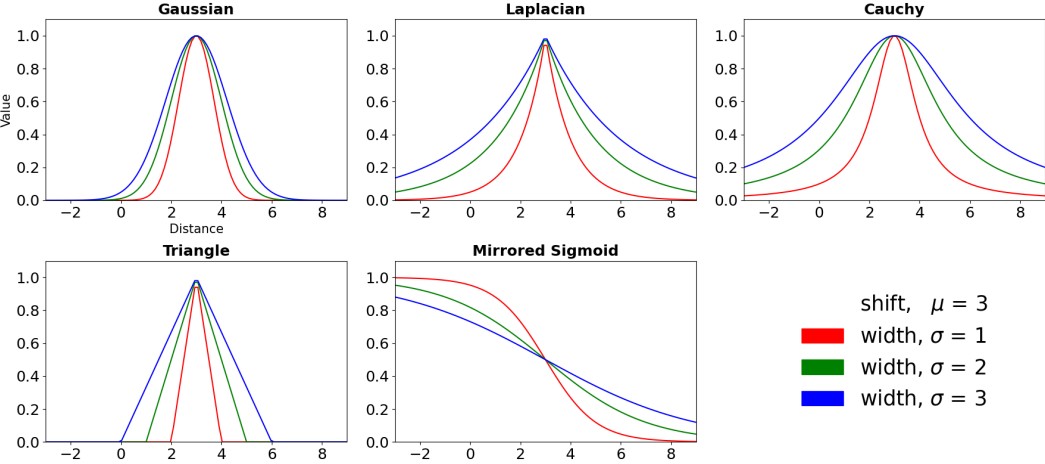

Figure 1: Visualization of different focus functions. Each function adjusts attention scores based on node distances, with unique characteristics influencing the attention distribution.

## 3.2 Computational Considerations

We make several adjustments to implement the focus mechanism efficiently and ensure stable learning. The equation 3 shows an updated formulation. We will explain the adjustments in the following.

$$\text{FocusedAttention}(Q, K, V, D') = \text{softmax}\left(\frac{QK^T}{\sqrt{d_k}} + \log\left(f\left(D', \mu', \sigma\right)\right) + W_{\text{self}}I\right)V \quad (3)$$

**Logarithmic Transformation:** Adding $\log(f(D_{ij}))$ inside the softmax is equivalent to multiplying $f(D_{ij})$ outside the softmax, as shown in Equation equation 4. This allows us to incorporate the focus mechanism into the attention scores using logarithms:

$$\text{softmax}\left(\frac{QK^T}{\sqrt{d_k}} + \log(f(D))\right) = \frac{f(D) \cdot \exp\left(\frac{QK^T}{\sqrt{d_k}}\right)}{\sum f(D) \cdot \exp\left(\frac{QK^T}{\sqrt{d_k}}\right)}. \quad (4)$$

Using the log of the focus function prevents numerical instabilities that could arise from very small values of $f(D_{ij})$, allowing for more stable computation and smoother gradients during backpropagation.

**Adding Self-loop:** To prevent the focus function from diminishing self-attention, especially when $\mu$ is large, we add an identity matrix scaled by a learnable weight to the distance matrix, as $D' = D + W_{\text{self}}I$. This ensures that each node maintains a significant attention score for itself.

**Masking to Prevent Unintended Global Nodes:** Zero-padded nodes might unintentionally create global nodes when padding graphs to create uniform batch sizes. To mitigate this, we implement masking that prevents attention and feed-forward network computations to and from these padded nodes.

**Masking Incorrect Distances:** In an adjacency matrix context, zero typically indicates no connection, making zero-padding a natural choice. However, this can lead to incorrect focusing values if the focus function interprets these zeros as valid distances. To address this issue, we assign a very large value (e.g., $10^6$) to padded nodes' distances in the matrix $D$, preventing any unintended influence from padded elements.

**Clamping $\mu$:** To ensure numerical stability and meaningful focusing behavior, we clamp $\mu$ to values greater than a minimum threshold $\mu' = \max(\mu_{\min}, \mu)$, where $\mu_{\min} = 0.5$. This prevents the focus function from becoming undefined or overly narrowed.

## 3.3 MoiréGT Architecture

The Moiré Graph Transformer (MoiréGT) architecture consists of multiple Moiré Layers that implement the focus mechanism (Figure 2).

**Input and Output Fully Connected Layers**: MoiréGT incorporates fully connected layers at both the input and output sides. The input layers preprocess raw node features, while the output layers map the transformed features to the desired output space. We also employed max pooling for the graph-level inference tasks to make embedding tensor before the output FC layers.

**Moiré Layer**: Each Moiré Layer consists of the focused attention mechanism integrated within the self-attention framework. The layer begins by projecting the input node features **X** into query, key, and value matrices. The focused attention is computed using the adjusted attention scores, incorporating the focus function as described in Equation equation 2. This allows the model to emphasize nodes within certain distance ranges dynamically.

**Feed-Forward Network and Residual Connections**: Following the attention mechanism, each layer includes a feed-forward network (FFN). Residual connections are employed around both the attention and FFN sublayers to prevent over-smoothing problems.

**Overall Architecture**: Multiple Moiré Layers are stacked to form the full MoiréGT model, allowing for hierarchical representation learning over the graph. Each layer can have multiple attention heads with different focus parameters, enabling the model to capture diverse structural patterns akin to the formation of moiré patterns.

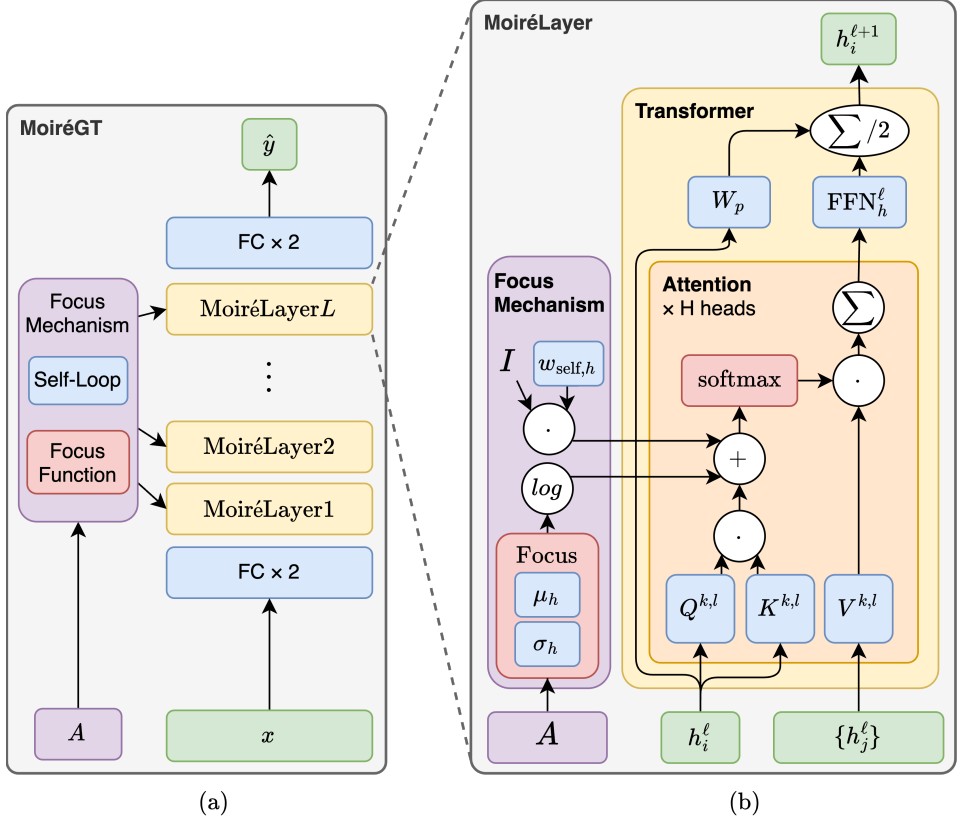

Figure 2: (a) Overview of the MoiréGT architecture, and (b) Detailed structure of a Moiré Layer with the focused attention mechanism.

### 3.4 Theoretical Foundation and Analysis

The Moiré Graph Transformer (MoiréGT) is inspired by moiré patterns, which arise when overlapping patterns with slight differences create complex interference. By integrating distance-based focus functions into the attention mechanism, MoiréGT implicitly encodes positional information, capturing complex graph structures without explicit positional encodings. The overlapping of multiple attention heads with varying focus parameters enables the model to capture local and global structural information effectively. For a detailed theoretical analysis, please refer to the Appendix.

## 4 Experiments and Results

We conducted a series of experiments to evaluate the performance and effectiveness of our MoiréGT model across various graph-based tasks and datasets. Our evaluation includes both inductive learning scenarios, covering 3D molecular graphs and 2D graph datasets.

### 4.1 Experimental Settings

We tested our model on several datasets, each with unique characteristics. We optimized the model's hyperparameters for each dataset, including the number of Moiré Layers, hidden

dimensions, attention heads, and learning rate. All experiments used the Gaussian focus function. The specific settings for each dataset are provided in Table 1.

**QM9:** A dataset of 134,000 small molecules with 3D geometric information, used for predicting molecular properties (Ramakrishnan et al., 2014).

**PCQM4Mv2:** A large-scale quantum chemistry dataset from the Open Graph Benchmark (OGB) containing over 3 million molecules with 3D structures (Hu et al., 2021).

**MNIST Superpixel:** A graph version of the MNIST dataset, where images are represented as graphs based on superpixels (Dwivedi et al., 2023).

Table 1: Hyperparameters for experimental datasets

| Dataset | Moiré Layers | Hidden Dim. | Heads | Learning Rate |
|---|---|---|---|---|
| PCQM4Mv2 | 8 | 512 | 16 | $2 \times 10^{-4}$ |
| QM9 | 13 | 256 | 32 | $5 \times 10^{-4}$ |
| MNIST Superpixel | 5 | 256 | 32 | $5 \times 10^{-4}$ |

## 4.2 Performance on 3D Graphs

Our first set of experiments focused on inductive learning tasks where the physical locations of nodes are provided, allowing us to calculate the adjacency matrix using actual physical distances between nodes. This is particularly relevant for molecular datasets like QM9 and PCQM4Mv2, where the 3D structures of molecules provide rich spatial information.

On the QM9 dataset, MoiréGT achieved a Mean Absolute Error (MAE) of 2.58 meV, significantly outperforming previous state-of-the-art models (Table 2). This demonstrates the model's effectiveness in capturing complex spatial relationships within molecular graphs without relying on explicit positional encodings.

Table 2: Results on QM9.

| Model | MAE (meV) ↓ |
|---|---|
| N-GramRF[1] | 10.37 |
| GROVER (large)[2] | 9.86 |
| GROVER (base)[2] | 9.84 |
| N-GramXGB[1] | 9.64 |
| D-MPNN[3] | 9.22 |
| PretrainGNN[4] | 9.22 |
| ChemRL-GEM[5] | 8.14 |
| Uni-Mol[6] | 4.67 |
| MoiréGT (Ours) | **2.58** |

**Bold** indicates the best, and underline indicates the second-best.
[1]Liu et al. (2019), [2]Rong et al. (2020), [3]Yang et al. (2019),
[4]Hu et al. (2019), [5]Fang et al. (2022), [6]Zhou et al. (2023)

Similarly, on the large-scale PCQM4Mv2 dataset, MoiréGT achieved an MAE of 46.3 meV on the validation set and 46.4 meV on the test-dev set (Table 3), surpassing all previous models, including those utilizing additional information from RDKit. This highlights the scalability and robustness of our approach to large and complex datasets.

## 4.3 Performance on 2D Graphs

To assess the generality of our approach, we evaluated MoiréGT on the MNIST Superpixel dataset, where node coordinates are provided in 2D space. We experimented with two settings: one where the distance matrix $D$ is computed using the Euclidean distances between nodes (MNIST-dist), and another where $D$ is based on the shortest path distances in the graph (MNIST-SPD).

Table 3: Results on PCQM4Mv2.

| Model | #Params | Val. MAE ↓ (meV) | Test-dev MAE ↓ (meV) |
|---|---|---|---|
| GINE[1]-VN[14] | 13.2M | 116.7 | - |
| GCN[2]-VN[14] | 4.9M | 115.3 | 115.2 |
| GIN[3]-VN[14] | 6.7M | 108.3 | 108.4 |
| DeeperGCN[4]-VN[14] | 25.5M | 102.1 | - |
| TokenGT[5] | 48.5M | 91.0 | 91.9 |
| GRPE[6] | 118.3M | 86.7 | 87.6 |
| Graphormer[7] | 47.1M | 86.4 | - |
| EGT[9] | 89.3M | 85.7 | 86.2 |
| GraphGPS[8] | 13.8M | 85.2 | 86.2 |
| Transformer-M[11] | 69M | 77.2 | 78.2 |
| GPS++[12] | 44.3M | 77.8 | 72.0 |
| TGT-At[15] | 203M | 68.6 | 69.8 |
| *Models below use RDKit | | | |
| GEM-2[10] | 32.1M | 79.3 | 80.6 |
| Uni-Mol+[13] | 77M | 69.3 | 70.5 |
| EGT[9] (2 Stage) | 189M | 69.0 | - |
| TGT[15] | 203M | 67.1 | 68.3 |
| MoiréGT (Ours) | 13.1M | **46.3** | **46.4** |

[1]Brossard et al. (2020), [2]Kipf & Welling (2016), [3]Xu et al. (2018),[4]Li et al. (2020),
[5]Kim et al. (2022), [6]Park et al. (2022), [7]Ying et al. (2021),[8]Rampášek et al. (2022),
[9]Hussain et al. (2022), [10]Liu et al. (2022),[11]Luo et al. (2022), [12]Masters et al. (2022),
[13]Lu et al. (2023),[14]Gilmer et al. (2017), [15]Hussain et al. (2024)

Our model achieved competitive results on MNIST-dist, with an accuracy of 97.79%, closely matching other state-of-the-art models (Table 4). However, the performance on MNIST-SPD was lower, with an accuracy of 94.72%.

Table 4: Results on MNIST Superpixel.

| Model | MNIST Accuracy (%) ↑ |
|---|---|
| GCN[1] | 90.71 |
| GIN[2] | 96.49 |
| Graphormer[3] | 97.91 |
| EGT[4] | **98.17** |
| GraphGPS[5] | 98.05 |
| MoiréGT-dist (Ours) | 97.79 |
| MoiréGT-SPD (Ours) | 94.72 |

[1]Kipf & Welling (2016), [2]Xu et al. (2018), [3]Ying et al. (2021),
[4]Hussain et al. (2022), [5]Rampášek et al. (2022)

The results indicate that MoiréGT effectively leverages physical node coordinates when available, as seen in the MNIST-dist setting. The reduced performance in the MNIST-SPD setting suggests that the model is less effective when relying solely on topological distances without explicit spatial coordinates.

## 4.4 ABLATION STUDY

We conducted an ablation study to evaluate the impact of different focus functions on MoiréGT's performance. The study tested Gaussian, Laplacian, Cauchy, Triangle, and MirroredSigmoid functions and a baseline model without any focus function.

As shown in Figure 3, the Gaussian focus function achieved the best performance on the QM9 dataset, with an MAE of 2.58 meV. The model without any focus function performed significantly worse, with an MAE of 29.12 meV, underscoring the importance of the focus mechanism. The Laplacian focus function failed to converge, suggesting that non-differentiable points in the Laplacian focus function may hinder training.

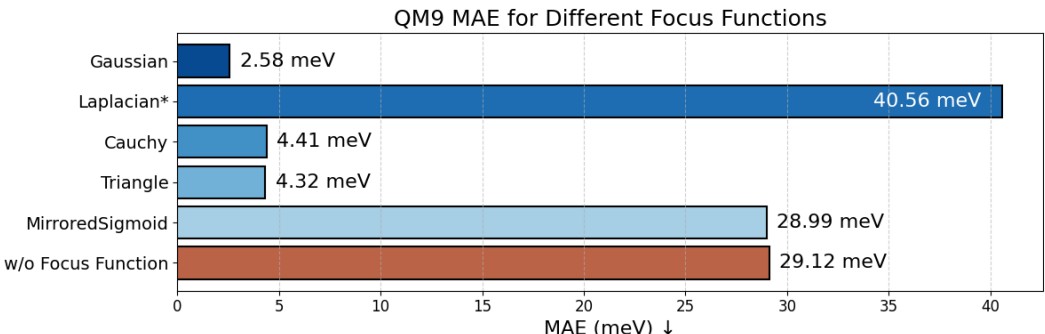

Figure 3: MAE of different focus functions on the QM9 dataset. Gaussian achieved the best MAE (2.58 meV), while the model without a focus function performed significantly worse. *The Laplacian function failed to converge, resulting in a high MAE (40.56 meV).

The ablation study highlights the effectiveness of the Gaussian focus function in capturing spatial relationships within the graph. The superior performance suggests that the smooth, bell-shaped curve of the Gaussian function provides an optimal balance between focusing on relevant nodes and maintaining gradient stability during training.

### 4.5 QUALITATIVE ANALYSIS OF FOCUS MECHANISM

To further understand the behavior of the focus mechanism, we analyzed the learned shift ($\mu$) and width ($\sigma$) parameters during training. Figure 4 shows how these parameters evolve over time.

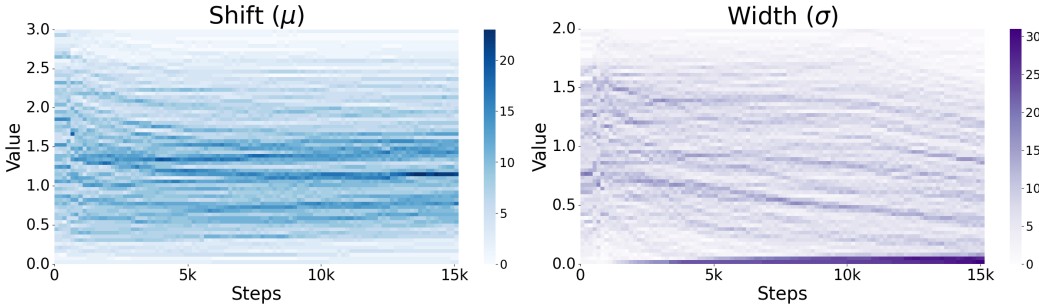

Figure 4: Evolution of mean shift ($\mu$) and width ($\sigma$) parameters during training. The dynamic adjustment of these parameters indicates the model's ability to adapt its focus to capture relevant structural information at different stages of learning.

The shifts and widths exhibit dynamic changes, indicating that the model adjusts its focus range and center as training progresses. This adaptability allows MoiréGT to capture varying scales of structural information, contributing to its strong performance on graph tasks.

## 5 CONCLUSION

We have introduced MoiréGT, a novel graph transformer architecture that eliminates the need for explicit positional encodings by leveraging a focused attention mechanism inspired

by moiré patterns. Our approach removes the reliance on global node tricks and complex positional encodings to infer graph-level properties and node positions, respectively.

By integrating self-loops and carefully designed focus functions, we stabilize the attention mechanism and gradients while effectively capturing structural information through implicit positional encoding. Our theoretical analysis demonstrates how the superposition of multiple focus functions can encode complex graph structures akin to the formation of moiré patterns.

Our experiments on 3D molecular datasets show that MoiréGT achieves state-of-the-art results, significantly outperforming existing models in predicting molecular properties. The model also demonstrates promising performance on 2D graphs.

While MoiréGT excels in tasks where physical node locations or meaningful distance measures are available, it underperforms in settings where such information is absent. This suggests that the current focus mechanism relies heavily on distance information, which may not be suitable for all types of graph data.

Future work could explore extending the focus mechanism to incorporate topological features or integrating effective positional encoding methods in these settings. We believe that further research can extend MoiréGT's applicability to a wider range of graph learning tasks.

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

# A APPENDIX

## A.1 DETAILED THEORETICAL FOUNDATION AND ANALYSIS

In this appendix, we provide a detailed theoretical foundation for the Moiré Graph Transformer (MoiréGT), demonstrating its mathematical superiority in encoding graph structural information without explicit positional encodings. We present two key proofs:

1. **Prove 1:** Optimal encoding of any given space can be achieved through a series of wavefunctions with different phase distortions. This encoding is analogous to the superposition of functions related to prime numbers and the Riemann Zeta function.

2. **Prove 2:** When focusing on local regions, any function can be approximated using Gaussian or other focus functions employed in our model.

### A.1.1 PROVE 1: OPTIMAL ENCODING VIA SUPERPOSITION OF WAVEFUNCTIONS

**Background on Wavefunction Superposition** In quantum mechanics and signal processing, any complex function can be represented as a superposition (sum) of simpler wavefunctions, such as sine and cosine functions with varying frequencies and phases. This principle is formalized in Fourier analysis, where a function $f(x)$ can be decomposed into its frequency components:

$$f(x) = \int_{-\infty}^{\infty} A(\omega)e^{i\omega x}, d\omega,$$ (5)

where $A(\omega)$ is the amplitude of the frequency component $\omega$, and $i$ is the imaginary unit.

**Encoding Space with Phase-Distorted Wavefunctions** Consider a graph where nodes are embedded in a continuous space. We aim to encode the positional relationships between nodes optimally. By employing a series of wavefunctions with different phase distortions, we can create interference patterns that uniquely encode spatial information.

Let $\psi_n(x)$ be a set of wavefunctions defined as:

$$\psi_n(x) = e^{i(k_n x + \phi_n)},$$ (6)

where $k_n$ is the wavevector (related to frequency), and $\phi_n$ is the phase distortion for the $n$-th wavefunction.

By superimposing these wavefunctions, we obtain:

$$\Psi(x) = \sum_{n=1}^{N} a_n \psi_n(x) = \sum_{n=1}^{N} a_n e^{i(k_n x + \phi_n)},$$ (7)

where $a_n$ are amplitude coefficients.

**Connection to Moiré Patterns** Moiré patterns emerge when two or more patterns with slight differences are overlaid, resulting in new patterns due to interference. In our context, the superposition of wavefunctions with different $k_n$ and $\phi_n$ values leads to interference patterns that can capture complex spatial relationships within the graph.

**Optimal Encoding and Prime Numbers** The distribution of prime numbers is connected to the zeros of the Riemann Zeta function $\zeta(s)$. The Riemann Zeta function can be expressed as an infinite series and has a deep connection with Fourier analysis through its Euler product representation:

$$\zeta(s) = \prod_{p,\text{prime}} \left(1 - \frac{1}{p^s}\right)^{-1}.$$ (8)

While the direct application of the Riemann Zeta function to graph encoding is non-trivial, we can draw an analogy by considering that the prime frequencies (analogous to prime numbers) in our wavefunction superposition contribute uniquely to the encoding, enhancing the distinctiveness of the positional representation.

**Proof of Optimal Encoding**   To prove that the superposition of wavefunctions with different phase distortions provides an optimal encoding, we rely on the completeness property of the Fourier basis. Any square-integrable function $f(x)$ can be approximated arbitrarily closely by a sum of wavefunctions:

$$f(x) \approx \sum_{n=1}^{N} a_n e^{i(k_n x + \phi_n)}, \tag{9}$$

where the coefficients $a_n$, frequencies $k_n$, and phases $\phi_n$ are chosen based on the function $f(x)$.

In our model, each attention head corresponds to a wavefunction with specific focus parameters (analogous to $k_n$ and $\phi_n$). By combining multiple attention heads, we effectively perform a superposition of focus functions, allowing us to approximate any desired encoding function over the node distances.

**Implications for MoiréGT**   The focused attention mechanism in MoiréGT, with multiple heads having different focus parameters, creates interference patterns similar to moiré patterns. This mechanism enables the model to implicitly encode positional information optimally, capturing both local and global structural patterns in the graph without explicit positional encodings.

### A.1.2   PROVE 2: LOCAL APPROXIMATION USING GAUSSIAN FOCUS FUNCTIONS

**Localization Property of Gaussian Functions**   Gaussian functions are widely used in signal processing and probability due to their localization properties. A Gaussian function centered at $\mu$ with width (standard deviation) $\sigma$ is defined as:

$$f(d) = \exp\left(-\frac{(d-\mu)^2}{2\sigma^2}\right). \tag{10}$$

This function peaks at $d = \mu$ and decays rapidly as $d$ moves away from $\mu$. The localization property allows Gaussians to approximate functions that are significant in a local region.

**Approximation of Local Functions**   Consider a smooth function $g(d)$ that is significant within a local region around $d = \mu$. We can approximate $g(d)$ using a Gaussian function by matching the function's value and derivatives at $d = \mu$.

**Taylor Series Expansion**   Let us expand $g(d)$ around $d = \mu$ using a Taylor series:

$$g(d) = g(\mu) + g'(\mu)(d-\mu) + \frac{1}{2}g''(\mu)(d-\mu)^2 + \cdots. \tag{11}$$

For small $(d - \mu)$, higher-order terms become negligible, and the function can be approximated by a quadratic form.

**Gaussian Approximation**   The Gaussian function inherently possesses a quadratic exponent, making it suitable for approximating functions locally. By adjusting the parameters $\mu$ and $\sigma$, the Gaussian function can be tuned to closely match $g(d)$ within the local region.

**Application in Focused Attention**   In the focused attention mechanism, we adjust the attention scores based on the distance between nodes using the Gaussian focus function:

$$\log(f(d)) = -\frac{(d - \mu)^2}{2\sigma^2}. \tag{12}$$

By incorporating this into the attention mechanism, we effectively emphasize nodes within a local region around $d = \mu$ and diminish the influence of nodes outside this region.

**Proof of Local Approximation**   Given that any smooth function can be locally approximated by a quadratic function, and the Gaussian function provides a quadratic form in its exponent, we can conclude that:

$$g(d) \approx A \exp\left(-\frac{(d - \mu)^2}{2\sigma^2}\right), \tag{13}$$

where $A$ is a scaling factor determined by $g(\mu)$.

**Implications for MoiréGT**   By employing Gaussian focus functions with learnable parameters $\mu$ and $\sigma$, MoiréGT can adaptively focus on relevant local structures within the graph. This allows the model to approximate any local function of node distances, effectively capturing local patterns and enhancing the expressive power of the attention mechanism.

### A.1.3   COMBINING MULTIPLE FOCUS FUNCTIONS

**Superposition Principle**   By using multiple attention heads, each with its own focus function parameters $(\mu_h, \sigma_h)$, we can combine the effects of different local approximations to capture more complex structures.

$$\text{Combined Focus} = \sum_{h=1}^{H} \alpha_h f_h(d), \tag{14}$$

where $\alpha_h$ are weighting coefficients, and $H$ is the number of attention heads.

**Universal Approximation**   The universal approximation theorem states that a sufficiently large weighted sum of activation functions can approximate any continuous function on a compact domain. Similarly, the superposition of multiple Gaussian functions with appropriate parameters can approximate any function over the node distances.

**Proof of Universal Approximation**   Given any continuous function $g(d)$ defined on a compact interval $[a, b]$, for any $\epsilon > 0$, there exists a finite sum of Gaussian functions such that:

$$\left| g(d) - \sum_{h=1}^{H} \alpha_h \exp\left(-\frac{(d - \mu_h)^2}{2\sigma_h^2}\right) \right| < \epsilon, \quad \forall d \in [a, b]. \tag{15}$$

This result follows from the properties of radial basis functions and their use in function approximation.

**Implications for MoiréGT**   By utilizing multiple attention heads with different Gaussian focus functions, MoiréGT can approximate complex positional encodings and capture intricate structural patterns in the graph. This capability contributes to the model's mathematical superiority in representing and learning from graph data without explicit positional encodings.

## A.2 Stability and Efficiency Considerations

**Numerical Stability**  In our implementation, we use the logarithm of the focus function inside the attention mechanism to maintain numerical stability:

$$\text{Attention}(Q, K, V) = \text{softmax}\left(\frac{QK^\top}{\sqrt{d_k}} + \log(f(D))\right) V. \tag{16}$$

This formulation prevents issues with very small values of $f(D)$ that could lead to numerical underflow.

**Computational Efficiency**  The focused attention mechanism avoids the need for explicit positional encodings or masking strategies, reducing computational overhead. By adjusting attention scores based on precomputed distance matrices and learnable focus parameters, we maintain the efficiency of the standard attention mechanism while enhancing its expressiveness.

## A.3 Conclusion of Theoretical Analysis

The theoretical proofs provided demonstrate that the Moiré Graph Transformer leverages fundamental principles from Fourier analysis and function approximation to achieve optimal encoding of graph structures. By employing a focused attention mechanism with multiple heads and learnable focus functions, the model captures both local and global structural information effectively.

These mathematical foundations underpin the model's superior performance in experiments, validating the design choices and highlighting the advantages of eliminating explicit positional encodings in favor of implicit, learnable mechanisms.

