# OpenReview forum: "Moiré Graph Transformer: Eliminating Positional Encoding with Focused Attention"
_ICLR.cc/2025/Conference — Submitted to ICLR 2025_

### Official Review · Reviewer_VqQk · 2024-10-28

**Soundness:** 2
**Presentation:** 3
**Contribution:** 2
**Rating:** 5
**Confidence:** 3

**Summary:**

This work proposes a GT model that eliminates the need for explicit graph positional encodings.

**Strengths:**

- This proposed graph transformer architecture does not need explicit positional encodings.
- The organization of the content is good.
- It is interesting to enable the model to capture diverse structural patterns akin to the formation of moiré patterns.

**Weaknesses:**

- The total time complexity of the proposed model could be analyzed and experimentally verified.
- The source codes and the used datasets should be provided to facilitate the reproducibility of this work. Please refer to https://anonymous.4open.science.
- Since this work presents a graph model, the widely-used graph benchmark datasets could be considered, such as the OGB datasets ogbn-arxiv, ogbn-mag, ogbn-products, ogbn-proteins, and ogbn-papers100M.

**Questions:**

Please see the weaknesses raised above.

---

### Official Review · Reviewer_vs57 · 2024-11-04

**Soundness:** 2
**Presentation:** 2
**Contribution:** 2
**Rating:** 3
**Confidence:** 4

**Summary:**

The paper introduces Moiré Graph Transformer (MoiréGT), a graph transformer architecture that eliminates the need for explicit positional encodings through a focused attention mechanism. The key innovation is a learnable focus function that adjusts attention scores based on node distances, allowing the model to implicitly encode structural information. The authors demonstrate that multiple attention heads with different focus parameters can capture complex structural patterns similar to moiré interference patterns. The model achieves state-of-the-art results on molecular property prediction tasks (QM9 and PCQM4Mv2 datasets) and shows competitive performance on 2D graph tasks.

**Strengths:**

1. The paper has a clear problem statement.
2. The focused attention mechanism can incorporate structural information without explicit positional encodings
3. No need of virtual nodes and complex positional encodings

**Weaknesses:**

1. Limited analysis of computational efficiency. There is no analysis of training time or memory usage compared to baseline methods.
2. The complexity analysis of the focus mechanism could be more detailed.
3. The implementation choices are not well reasoned. The choice of focus function forms (Gaussian, Laplacian, etc) seems somewhat arbitrary. It is not clear why or how the author choose them.
4.Some technical implementation details, like the numerical stability benefits of the logarithmic transformation, need better explanation and justification

**Questions:**

1. How does the computational complexity of MoiréGT compare to other graph transformers, particularly in terms of memory usage and training time?
2. Regarding numerical stability, the paper claims that using log(f(D)) inside the softmax prevents numerical instabilities. Could you explain:

- What specific numerical instabilities would occur without the log transformation?
- Why is it mathematically equivalent to multiply f(D) outside the softmax versus adding log(f(D)) inside it?
- How does this affect the gradient computation during backpropagation?


3. Can you provide more insight into why the Laplacian focus function failed to converge? Are there specific properties of the function that make it unsuitable?
4. How does the model's performance scale with graph size? Are there limitations on the maximum graph size that can be effectively processed?

---

### Official Review · Reviewer_NHBd · 2024-11-04

**Soundness:** 2
**Presentation:** 2
**Contribution:** 2
**Rating:** 5
**Confidence:** 4

**Summary:**

This paper introduces a new focused attention mechanism for graph transformers that eliminates the need for explicit positional encodings. By leveraging a learnable distance-based focus function, the model can adjust attention scores based on the spatial relationships between nodes, effectively capturing structural information within a graph. Experiments show that the proposed model obtains better performance than existing methods on 3D and 2D graph benchmarks.

**Strengths:**

- This model captures the structural information through a parameterized focus function and thus reduces the reliance on complicated positional encoding
- The proposed model shows good performance on 3D and 2D graph benchmarks and outperforms existing methods.

**Weaknesses:**

- **Title and model name**: I think the description of moiré patterns should be put in the front part of the main body to help readers better understand the title and the model name.
- **Not enough benchmarks**: The model is only evaluated on two 3D graph datasets and one 2D graph dataset, which seems not enough compared with the baseline papers.
- **Initialization and interpretation of focus function**: This paper didn’t describe how the learnable parameters (i.e., shift and width) of the focus function are initialized and how the initialization affects the performance. I would expect it to be important since it gives the model a prior before training on which nodes could be important. Besides, Figure 4 only shows the change of shift and width over time. What does the intensity of the color mean? It would be good to interpret the learned focus function and relate it to the particular properties of the dataset.
- **Related work**: [1] shares a similar perspective (e.g., removing PE and learning a distance-based filter) and should be discussed.

[1] Recurrent Distance Filtering for Graph Representation Learning. ICML'24.

**Questions:**

Please see Weaknesses

---

### Official Review · Reviewer_3Eu1 · 2024-11-04

**Soundness:** 2
**Presentation:** 2
**Contribution:** 2
**Rating:** 3
**Confidence:** 4

**Summary:**

The paper proposes Moire Graph Transformer (MoireGT) equipped with the newly proposed focus attention mechanism. Existing graph Transformers inject graph structural information into graph Transformers with two approaches: positional encoding and masking approaches. But, designing encodings that uniquely represent topological positions while preserving invariance properties is difficult, and masking approaches have a problem where they restrict the scope of the attention compared to the self-attention. To address this issue, the paper proposes the focus attention that focuses on specific ranges of nodes without losing the capability of capturing global receptive field of attention mechanisms.

**Strengths:**

- The paper deals with the design of graph Transformers, which is an important research topic in the graph representation learning domain. Transformers have been successfully applied in various areas, but it is hard to say that they are effective in representing graphs due to the difficulty of encoding graph structural information. This paper proposes the focused attention mechanism to capture the structural context of graphs.

**Weaknesses:**

- It would be better if the paper included more related works in Section 2.
- Could you please explain the meaning of a sentence, "While positional encodings have been employed to inject structural information (Dwivedi et al., 2021), designing encodings that uniquely and meaningfully represent topological positions while preserving invariance properties is challenging."?
- It would be better if the paper provided more analysis explaining how the proposed MoireGT performs well. The paper presents two analyses: the performance of MoireGT according to the focus function and the qualitative analysis of the focus mechanism. But, both analyses do not explain why and how MoireGT works well in understanding graph-structured data. In particular, Figure 4 simply shows that the shift $\mu$ and width $\sigma$ evolve over time, and that's it. From Figure 4, I find that the width tends to become zero as the number of steps becomes bigger. I think that it is because each node prefers to attend the nodes having $\mu$ distance. Could you discuss this phenomenon?
- From my knowledge, QM9 dataset has multiple subtasks. But, the paper shows the experimental result of a single task.

**Questions:**

Please refer to weaknesses section.

---

### Meta-Review · Area_Chair_Svtj · 2024-12-19

**Metareview:**

**Summary:**

The paper proposes Moire Graph Transformer equipped with a new attention mechanism. Prior graph Transformers incorporate graph structural information into Transformers with either positional encoding or an attention mask. However, both methods have limitations such as difficulty in representing the original topology while preserving invariance. To address this, the authors proposed a new attention mechanism that focuses more on a specific range of nodes.

**Strengths:**

- **Problem identification.** This paper clear introduces the main problem; the limitation of prior graph transformers in incorporating graph topology in attention mechanisms while preserving invariance without sacrificing expressive power of attention mechanisms.

**Weaknesses:**

- **Weak experiments/insufficient benchmarks.** This paper provides only a few experimental results on 2D/3D graph datasets. More extensive experiments are needed to compare the proposed method with stronger baselines.
- **Computational efficiency.**  No analysis of training time or memory usage is provided compared to baseline methods.
- **Title and model name.**  As a reviewer explicitly mentioned, the title/model name “moiré” patterns should be introduced to help readers better associate the name with the main contributions.

**Main reasons:**

The weak experimental results and limited comparison with strong baselines are the main reasons why this paper is recommended for rejection. Additionally, no author feedback was provided. Therefore, none of the concerns raised by the reviewers has been addressed.

**Additional Comments On Reviewer Discussion:**

No author feedback has been provided. Hence, none of the concerns raised by reviewers has been addressed. As mentioned above, weak experimental results and no comparison with stronger baselines on common benchmarks make it difficult to evaluate the contributions.

---

### Decision · Program_Chairs · 2025-01-22

Reject